# Experimental Study and Bearing Capacity Analysis of Retrofitted Built-Up Steel Angle Members under Axial Compression

**Junxin Li [1], Xiaohong Wu [2], Bin Yang [1], Jiaqi Wang [1] and Qing Sun [1,*]**

[1] Department of Civil Engineering, Xi'an Jiaotong University, Xi'an 710049, China; lijunxin@stu.xjtu.edu.cn (J.L.)

[2] School of Aerospace, Xi'an Jiaotong University, Xi'an 710049, China

* Correspondence: sunq@mail.xjtu.edu.cn

**Abstract:** Against the update of load design standards and requirement of long-term service, many latticed towers using steel angles subjected to gradual performance degradation must be retrofitted for bearing further social functions, e.g., electric power transmission and fifth-generation mobile communication. Therefore, this paper proposed a non-destructive reinforcement method for steel angles by avoiding unnecessary new construction or complex reinforcement procedure. The non-destructive reinforcement of a steel angle is composed of a steel angle, reinforcement plate, and fixture. Standardized fixtures are used to connect the reinforcement plate and steel angle to achieve steel angle reinforcement. In retrofitting tests, built-up steel angle members were loaded under axial compression, in which the failure mode and reinforcement effect of key parameters (e.g., clamp type, slenderness ratio, and clamp distance) were analyzed and compared, where a significant reinforcement effect was obtained with the capacity increment within 39~174%; the clamp types and clamp distance had a slight effect on bearing capacity; and the proposed reinforcement method was more effective for slender members. Based on the mechanical mechanism analysis and failure mode, the accuracy of the design method for calculating the bearing capacity of those built-up steel angle members was suggested and verified, in which a simplified mechanical model for the flexural-buckling mode was developed. The design method based on AISC360-16 agreed well with the test result and could be effectively used for calculating the flexural–torsional bearing capacity of those built-up steel angles. This study can provide a valuable reference for the design and application of non-destructive reinforcement of angle steel towers.

**Keywords:** non-destructive reinforcement; built-up steel angles; failure mode; bearing capacity; design method

## 1. Introduction

Latticed steel angle towers are widely used in lifeline engineering structures such as communication towers, transmission towers, and television towers due to their advantages of regular structural form, uniform stiffness changes, and simple processing technology [1–5]. As an important component of large and complex lifeline systems, safety issues directly affect the production and construction of the country and the living order of the people. There are approximately 2 million communication towers in China, most of which are lattice steel angle towers. Due to extreme weather conditions such as strong winds, ice cover, and earthquakes, insufficient consideration is given to design wind loads, especially with the emergence of new technologies such as 5G, which require a large number of additional equipment to be installed on existing communication towers [4], and other reasons [6–12], resulting in loads exceeding the bearing capacity of existing steel angle towers and causing structural damage or even collapse. Due to the fact that the cost of constructing new steel angle towers is much higher than the cost of strengthening existing steel angle towers,

telecom operators attach great importance to strengthening them in order to achieve higher economic benefits [13–16].

At present, there are three main types of reinforcement methods for lattice towers in engineering: one is to increase the transverse diaphragm or auxiliary materials, reduce the calculated length of the members, and improve the buckling bearing capacity. For example, Albermani et al. [17] added a series of partition supports in the middle of slender members, which can achieve significant load-bearing capacity improvement. Xie and Sun [18] strengthened the diagonal steel tower by adding diaphragms, and the overall bearing capacity of the tower was increased by about 18% after reinforcement. The second is to paste a fiber-reinforced polymer layer on the original components to improve the cross-sectional strength. For example, Lanier [19] proposed a renovation plan that utilizes high-modulus carbon fiber polymer as a reinforcement mechanism for communication towers. After reinforcement, the strength and stiffness of the cross-section were increased by 20–50%. Balagopal et al. [20] studied using fiberglass-reinforced plastic (GFRP) plates to reinforce existing communication towers. Yoresta et al. [21] proposed an unbonded CFRP reinforcement method to improve the axial bearing capacity of components. The third is to improve the structural bearing capacity by connecting other components in parallel based on the existing members. For example, Mills et al. [22] conducted a study on the reinforcement method of a cross-shaped double steel angle; in the experiment, the bearing capacity of the reinforced tower increased by 54–105%, and the reinforcement effect was significant. Yan et al. [23] examined three different forms of cross-shaped steel angle reinforcement components (one-way connection with filled plates, staggered connection with filled plates, and steel angle connection), and the results showed that the angle steel connection method was the best. Mills et al. [24] studied the reinforcement methods of cross joints and splice joints on the legs of transmission towers. Trovato et al. [25] designed a reinforcement method that prevents lateral torsion and weak axial buckling of single-angle section members through a casing and achieved good results overall. Lu et al. [26,27] conducted an experimental and numerical study on the reinforcement method of anisotropic cross-shaped components in the legs of lattice transmission or communication towers, effectively improving the overall strength of the structure. Ananthi et al. [28] conducted experimental research on the welding and fastening of back-to-back combination unequal-leg cold-bending steel angles under axial compression, and the results showed that the predicted bearing capacity of unequal leg combination angle steel using the direct strength method was about 7% lower. Dar et al. [29] conducted a study on composite columns made of four limb steel angles, and the results showed that the slenderness ratio, connection form, and end plate width have a significant impact on the mechanical behavior of the columns. Overall, the third reinforcement method is the most effective and widely used. However, this reinforcement method requires positioning, drilling, and welding on the existing components, and the construction is complex. Generally, the latticed towers are usually located in remote deserts and mountainous areas, and the construction environment is harsh, making it difficult to implement.

Therefore, in this study, it is recommended to use a combination method for non-destructive reinforcement of steel angle. Furthermore, compression tests were conducted on composite steel angle members to verify the synergistic mechanism in reinforcement behavior, and a simplified prediction method was finally established to calculate the buckling bearing capacity of these new composite steel angle members. This study can provide a valuable reference for the design and application of non-destructive reinforcement of steel angle towers.

## 2. Retrofitting Test on Steel Angles

This paper provides a new method of steel angle reinforcement for prefabricated composite structures. Compared to traditional steel angle reinforcement methods, it has many advantages, such as convenient installation and construction, no need for welding, drilling, etc. However, its mechanical performance is not yet clear, and there is no applicable

concise calculation method for bearing capacity, making it difficult to apply it to engineering on a large scale. Therefore, the main purpose of this paper is to obtain the bearing capacity of steel angle-reinforced components, analyze the stress mechanism of steel angle-reinforced components, establish a mechanical model of steel angle-reinforced components, and provide a basis for establishing a concise calculation formula for the bearing capacity of retrofitted built-up steel angle members.

### 2.1. Specimen Design

The proposed schematic diagram of the steel angle reinforcement method is shown in Figure 1. The steel angle reinforcement composite component consists of a steel angle, reinforcement plate, and fixture. The reinforcement plate is placed on the inner side of the weak axis of the steel angle, and the reinforcement plate is chamfered to ensure full contact with the inner side of the steel angle. There are two rows of bolt holes on the fixture and reinforcement plate, which are connected to the fixture through high-strength bolts of grade 8.8. The tightening torque should strictly meet the requirements to ensure that the reinforcement plate and steel angle can work together. The width of the reinforcement plate can be adjusted within a certain range to meet different bearing capacities and construction requirements. In the experiment of built-up steel angle members, the length of the reinforcement plate is slightly smaller than the length of the steel angle, which avoids direct compression of the reinforcement plate, facilitates the installation of the reinforcement plate, and is consistent with the actual engineering reinforcement method.

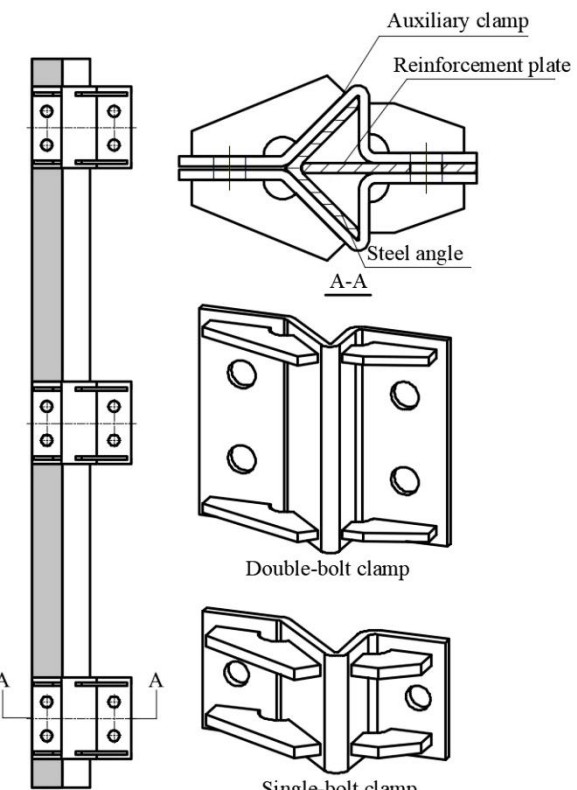

**Figure 1.** Retrofitting method.

In order to study the actual effect of a new prefabricated reinforcement form and analyze its reinforcement mechanism, five commonly used steel angle specifications (L80 × 7, L100 × 8, L125 × 10, L160 × 12, L180 × 14) in steel towers were selected for component testing conduct comparative experiments between reinforced and unreinforced specimens, taking into account factors such as fixture spacing, slenderness ratio, and fixture size in the design of each set of test specifications to study the applicability of this reinforcement

form and provide a basis for optimizing the reinforcement form. The test components are divided into two types: reinforced components and unreinforced components. The reinforced component is numbered LR, the unreinforced component is numbered LS, and there are five different steel angle specifications, L80 × 7, L100 × 8, L125 × 10, L160 × 12, L180 × 14 are numbered 1–5 in sequence, and the specimen design is shown in Figure 2. For ease of loading, end plates are welded at both ends of the steel angle, and a 5 mm gap is left between the reinforcement plate and the end plate, as shown in Figure 2a. This test designs two types of fixtures, namely single-row bolt form and double-row bolt form, and verifies the influence of different fixture sizes on the bearing capacity of reinforced specimens through comparative experiments. The partially reinforced plate between the end plate and the last fixture cannot work together with the steel angle due to insufficient constraints. Therefore, this is the weak section of the reinforced specimen. If the weak area is too long, the reinforced specimen will first experience local buckling here. To avoid local buckling, this article adds an additional pair of fixtures, such as LR2-2-2, to the ends of some specimens with longer weak areas to constrain the two ends of the reinforced plate and ensure that it and the steel angle can synergistically bear the force.

The thickness of the reinforcement plate in the reinforcement component is equal to the thickness of the steel angle limb, and the width of the reinforcement plate is the steel angle limb width/$\sqrt{2}$ + 80 mm (80 mm is about 3d, which is the minimum bolt edge distance × 2). The bolt adopts grade 8.8 M24, and the pre-tightening torque of the bolt is based on the requirements of the "Technical Regulations for High Strength Bolt Connection of Steel Structures". The pre-tightening force value is 175 kN, and the summary of specimen size is shown in Table 1.

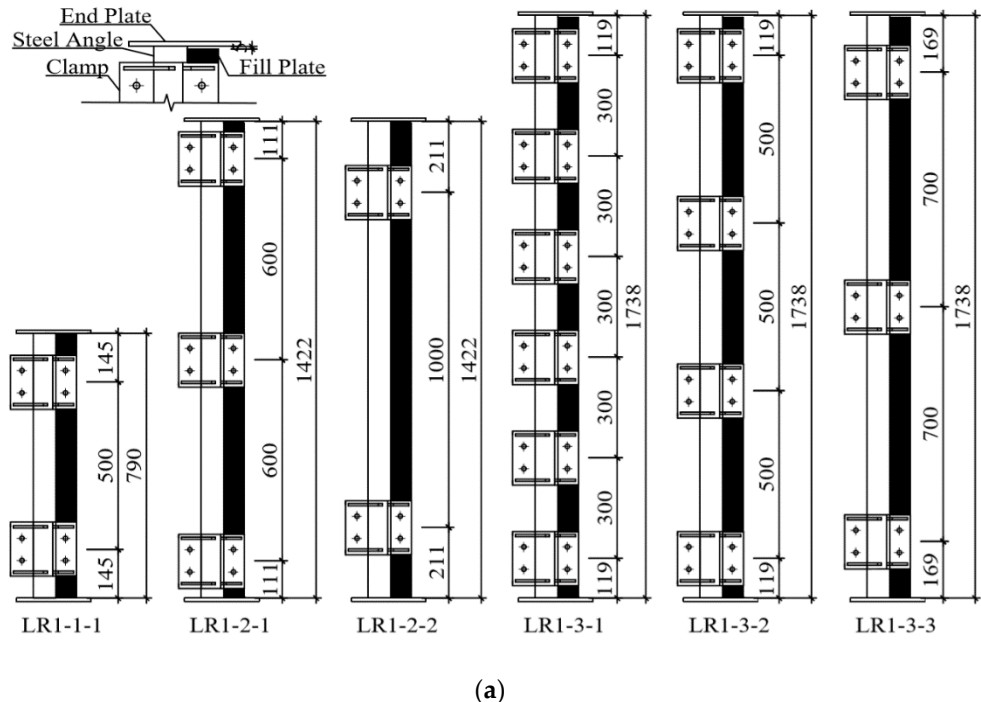

(**a**)

**Figure 2.** *Cont.*

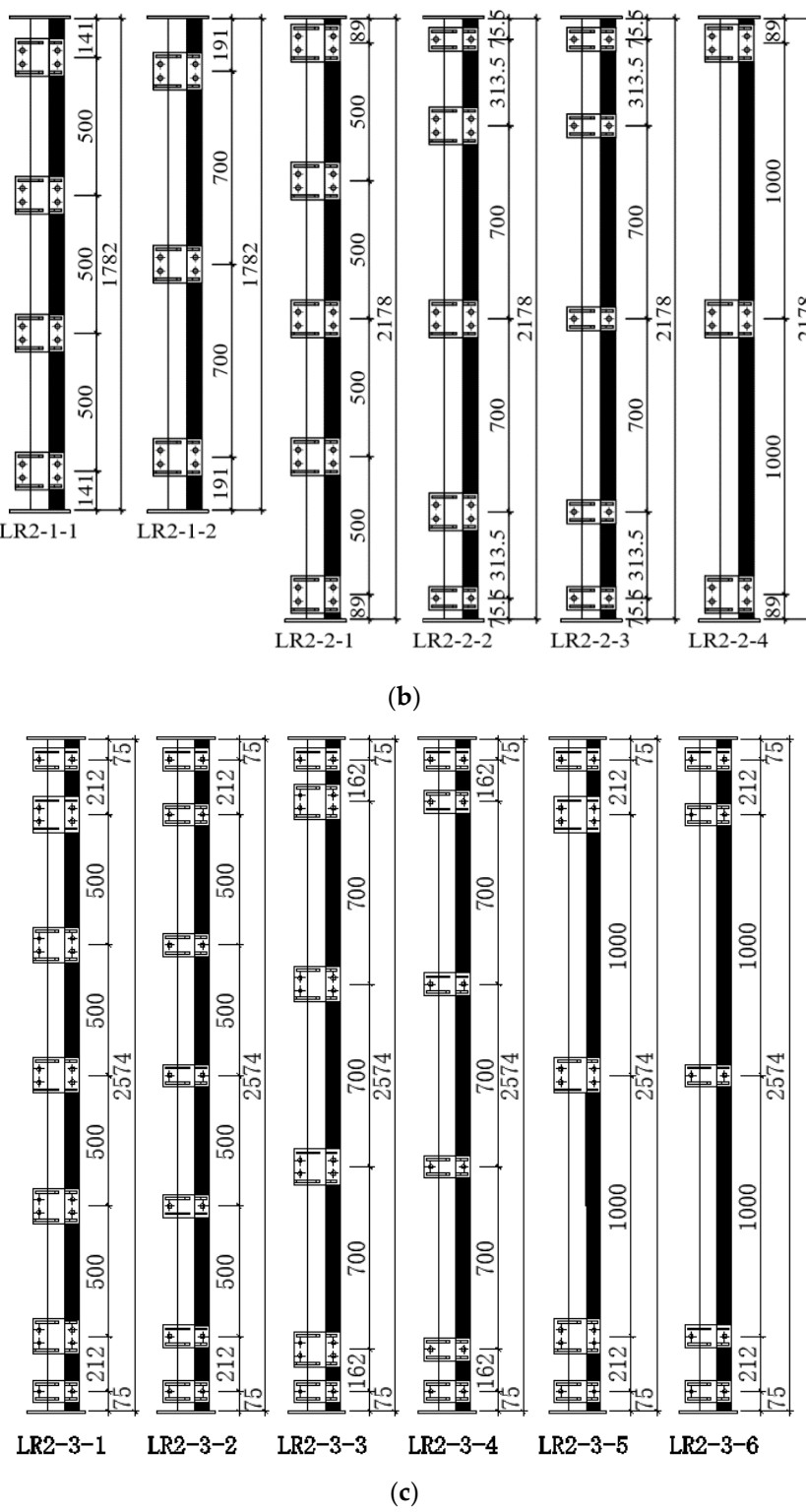

(**b**)

(**c**)

**Figure 2.** *Cont.*

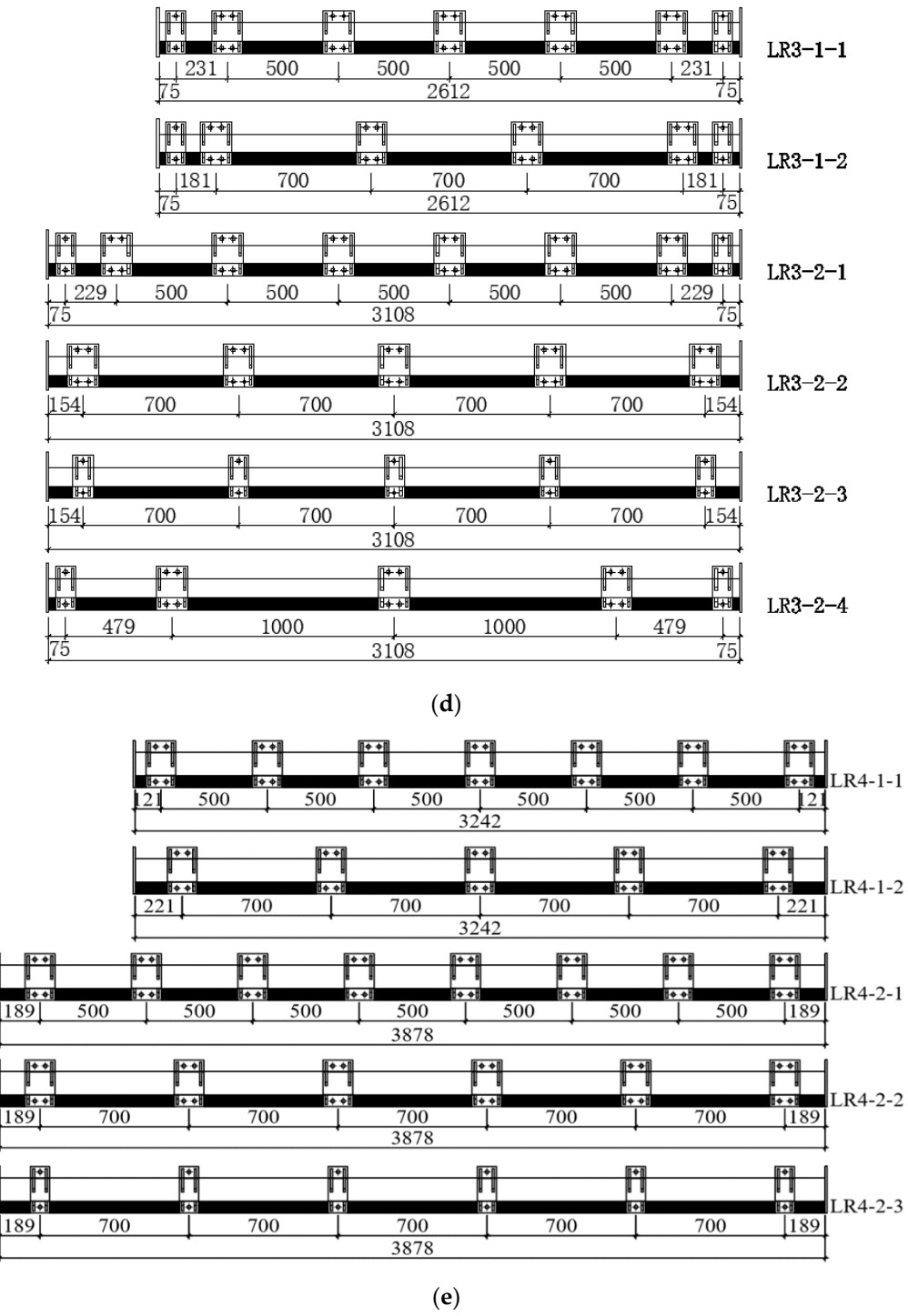

(**d**)

(**e**)

**Figure 2.** *Cont.*

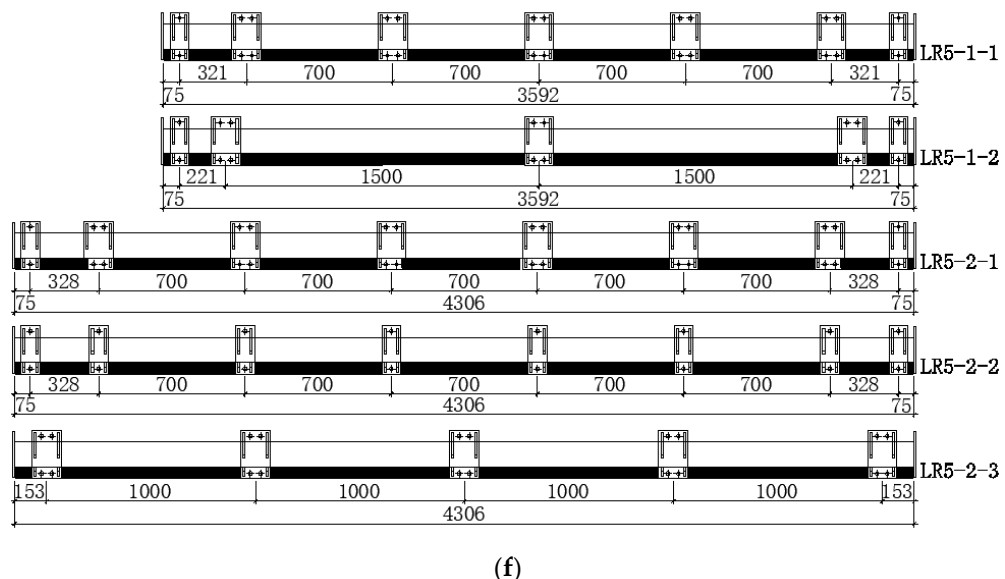

(**f**)

**Figure 2.** Specimen design. (**a**) Specimens of L80 × 7; (**b**) Specimens of L100 × 8 group I; (**c**) Specimens of L100 × 8 group II; (**d**) Specimens of L125 × 10; (**e**) Specimens of L160 × 12; (**f**) Specimens of L180 × 14.

**Table 1.** Summary of the tested built-up steel angles.

| Type | No. | Slenderness Ratio | Width of Reinforcement Plate/mm | Length of Reinforcement Plate/mm | Retrofitting Distance/mm | Clamp Type |
|------|-----|-------------------|----------------------------------|-----------------------------------|---------------------------|------------|
| | LS1-1-1 | 50 | \ | \ | \ | \ |
| | LR1-1-1 | 50 | 137 | 780 | 500 | Double-bolt |
| | LS1-2-1 | 90 | \ | \ | \ | \ |
| | LR1-2-1 | 90 | 137 | 1412 | 600 | Double-bolt |
| | LR1-2-2 | 90 | 137 | 1412 | 1000 | Double-bolt |
| L80 × 7 | LS1-3-1 | 110 | \ | \ | \ | \ |
| | LR1-3-1 | 110 | 137 | 1728 | 300 | Double-bolt |
| | LR1-3-2 | 110 | 137 | 1728 | 500 | Double-bolt |
| | LR1-3-3 | 110 | 137 | 1728 | 700 | Double-bolt |
| | LS2-1-1 | 90 | \ | \ | \ | \ |
| | LR2-1-1 | 90 | 151 | 1772 | 500 | Double-bolt |
| | LR2-1-2 | 90 | 151 | 1772 | 700 | Double-bolt |
| | LS2-2-1 | 110 | \ | \ | \ | \ |
| | LR2-2-1 | 110 | 151 | 2168 | 500 | Double-bolt |
| | LR2-2-2 | 110 | 151 | 2168 | 700 | Double-bolt |
| | LR2-2-3 | 110 | 151 | 2168 | 700 | Single-bolt |
| L100 × 8 | LR2-2-4 | 110 | 151 | 2168 | 1000 | Double-bolt |
| | LS2-3-1 | 130 | \ | \ | \ | \ |
| | LR2-3-1 | 130 | 151 | 2564 | 500 | Double-bolt |
| | LR2-3-2 | 130 | 151 | 2564 | 500 | Single-bolt |
| | LR2-3-3 | 130 | 151 | 2564 | 700 | Double-bolt |
| | LR2-3-4 | 130 | 151 | 2564 | 700 | Single-bolt |
| | LR2-3-5 | 130 | 151 | 2564 | 1000 | Double-bolt |
| | LR2-3-6 | 130 | 151 | 2564 | 1000 | Single-bolt |

**Table 1.** *Cont.*

| Type | No. | Slenderness Ratio | Width of Reinforcement Plate/mm | Length of Reinforcement Plate/mm | Retrofitting Distance/mm | Clamp Type |
|---|---|---|---|---|---|---|
| L125 × 10 | LS3-1-1 | 90 | \ | \ | \ | \ |
| | LR3-1-1 | 90 | 168 | 2212 | 500 | Double-bolt |
| | LR3-1-2 | 90 | 168 | 2212 | 700 | Double-bolt |
| | LS3-2-1 | 110 | \ | \ | \ | \ |
| | LR3-2-1 | 110 | 168 | 2718 | 500 | Double-bolt |
| | LR3-2-2 | 110 | 168 | 2718 | 700 | Double-bolt |
| | LR3-2-3 | 110 | 168 | 2718 | 700 | Single-bolt |
| | LR3-2-4 | 110 | 168 | 2718 | 1000 | Double-bolt |
| L160 × 12 | LS4-1-1 | 90 | \ | \ | \ | \ |
| | LR4-1-1 | 90 | 193 | 2852 | 500 | Double-bolt |
| | LR4-1-2 | 90 | 193 | 2852 | 700 | Double-bolt |
| | LS4-2-1 | 110 | \ | \ | \ | \ |
| | LR4-2-1 | 110 | 193 | 3488 | 500 | Double-bolt |
| | LR4-2-2 | 110 | 193 | 3488 | 700 | Double-bolt |
| | LR4-2-3 | 110 | 193 | 3488 | 700 | Single-bolt |
| L180 × 14 | LS5-1-1 | 90 | \ | \ | \ | \ |
| | LR5-1-1 | 90 | 207 | 3203 | 700 | Double-bolt |
| | LR5-1-2 | 90 | 207 | 3203 | 1500 | Double-bolt |
| | LS5-2-1 | 110 | \ | \ | \ | \ |
| | LR5-2-1 | 110 | 207 | 3917 | 700 | Double-bolt |
| | LR5-2-2 | 110 | 207 | 3917 | 700 | Single-bolt |
| | LR5-2-3 | 110 | 207 | 3917 | 1000 | Double-bolt |

## 2.2. Material Property

At the same time, the coupon test was conducted by testing the reserved same batch of steel angles to obtain the actual performance. The test results of all material specimens are listed in Table 2.

**Table 2.** Material properties of steel angle members.

| Type | Yield Strength $(f_y)$/MPa | Ultimate Strength $(f_u)$/MPa | Elastic Modulus $(E_s)$/GPa | Poisson's Ratio | Elongation/% |
|---|---|---|---|---|---|
| L80 × 7 | 396 | 535 | 199.6 | 0.282 | 27.4 |
| L100 × 8 | 378 | 527 | 205.1 | 0.289 | 27.3 |
| L125 × 10 | 384 | 531 | 205.3 | 0.287 | 26.6 |
| L160 × 12 | 383 | 539 | 201.3 | 0.296 | 27.2 |
| L180 × 14 | 394 | 540 | 207.5 | 0.298 | 28.9 |

## 2.3. Test Program

The loading points of all specimens (reinforced and unreinforced) in this test are located at the centroid of the steel angle, and the steel plates used for reinforcement do not directly bear the load. To meet the boundary conditions of the hinge joint at both ends, the test piece adopts bidirectional knife hinge supports at both ends. The hinge supports are placed in the same direction as the symmetrical axis of the steel angle, ensuring that the end of the test piece can rotate freely around the $x_0$ and $y_0$ axes of the steel angle, as shown in Figure 3.

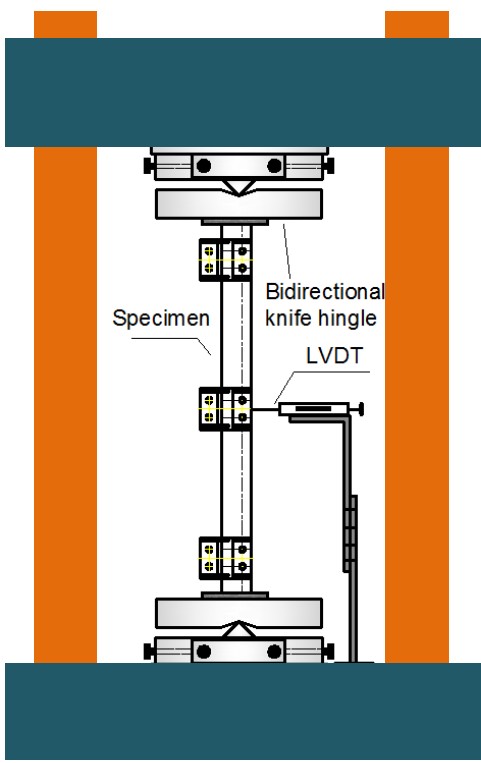

**Figure 3.** Test setup.

Static loading is carried out by a hydraulic testing machine. Prior to the test, a numerical analysis of the components is carried out to obtain simulated values of the ultimate bearing capacity of the components, which are used as estimated values for the test loading. The loading system is divided into two stages. The first stage adopts a vertical force control loading method, which is applied step by step. Each level of load takes 5% of the estimated ultimate load until it reaches 80% of the estimated ultimate load and then moves on to the second stage. The second stage adopts a vertical displacement-controlled loading method with a loading rate of 0.1 mm/min. The failure sign is that after the load–displacement curve of the specimen crosses the peak point, the bearing capacity decreases to 65% of the peak load, and then the loading ends.

In this paper, a resistance strain gauge is used to measure the strain variation in steel angle and reinforced plate during loading. For unreinforced steel angle specimens, the measurement points are distributed at the end of the specimen, and the mid-span section, with four measurement points arranged in each section, as shown in Figure 4a. For reinforced steel angle specimens, the measurement points are distributed at the mid-span and fixture nodes, with eight measurement points arranged in each section, of which four are on the displacement steel angle, and the others in Figure 4b are on the reinforced plate, as shown in Figure 4.

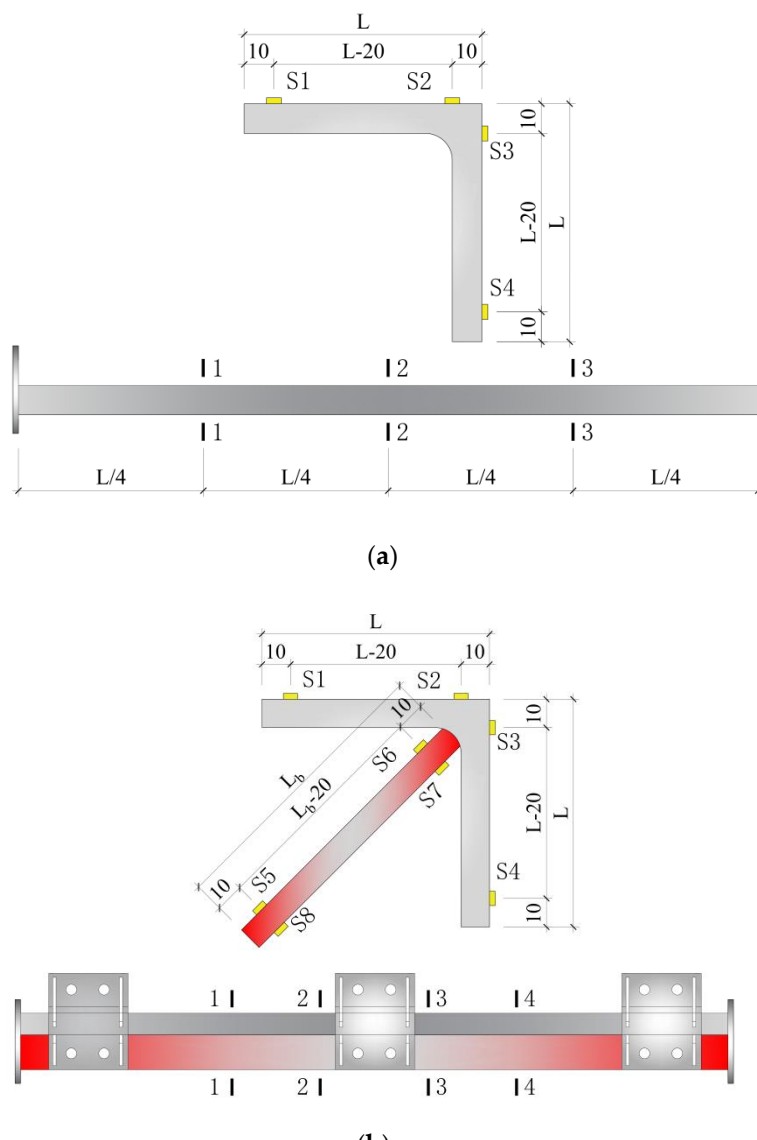

**Figure 4.** Strain measurement. (**a**) Measuring points of steel angles; (**b**) Measuring points of retrofitted steel angles.

### 2.4. Failure Mode

Both unreinforced steel angles and reinforced steel angles belong to uniaxial symmetric members, and the centroid of the cross-section is not at the same point as the shear center. Under axial compression, bending instability around the asymmetric axis or bending torsional instability around the symmetric axis may occur. In the experiment, all unreinforced steel angles experienced bending instability, and the reinforced steel angles exhibited two failure modes: bending and bending torsional instability, which are now introduced separately.

#### 2.4.1. Flexural–Buckling Mode

LR2, LR3, LR4, and LR5 groups of specimens all experienced bending instability around the asymmetric axis, and after buckling, all specimens underwent compression at the back of the limb and tension at the tip of the limb, as shown in Figure 5. There are two main reasons for the difference in failure modes: firstly, the LR2~LR5 group steel angle has a larger size, so the width of the reinforcement plate is smaller compared to the steel angle, and the reinforcement effect cannot change the original strength axis like the LR1

group specimen. Secondly, there is a certain installation gap between the steel angle and the reinforcement plate. When the steel angle flexes due to vertical load, it needs to overcome the installation gap first before it can fully contact the reinforcement plate. Therefore, the deformation of the steel angle always bends towards the limb tip side first, which increases the initial bending of the specimen in the asymmetric axis direction.

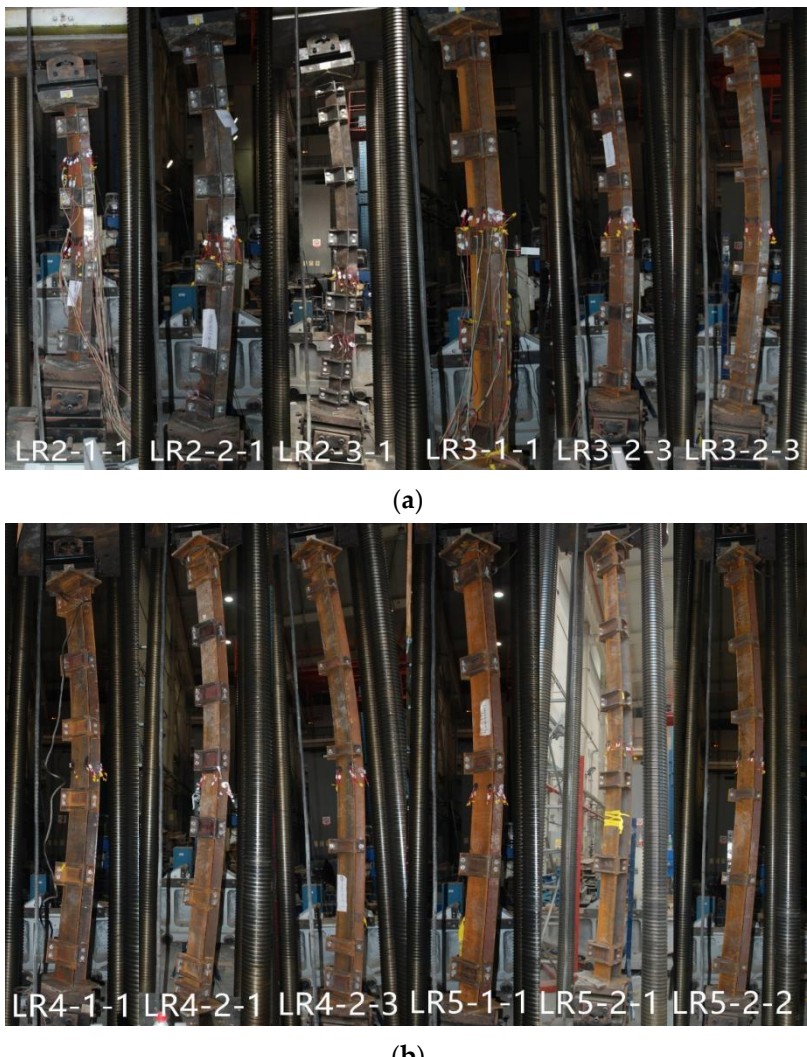

**Figure 5.** Flexural-buckling failure mode. (**a**) Specimens of LR2 and LR3; (**b**) Specimens of LR4 and LR5.

2.4.2. Flexural–Torsional Buckling Mode

The test results indicate that under axial load, all LR1 specimens exhibit bending and torsional instability failure patterns around the axis of symmetry. In the experiment, the spacing between fixtures has no significant effect on the failure mode, so only one type is listed for each group of specimens with the same slenderness ratio. As shown in Figure 6, the lateral bending deformation of the specimens is generated around the symmetrical axis of the steel angle, and rotation of the end hinge can be observed. After reinforcement, the original weak axis of the steel angle is strengthened, and the bending slenderness ratio is increased $\lambda x < \lambda yz$.

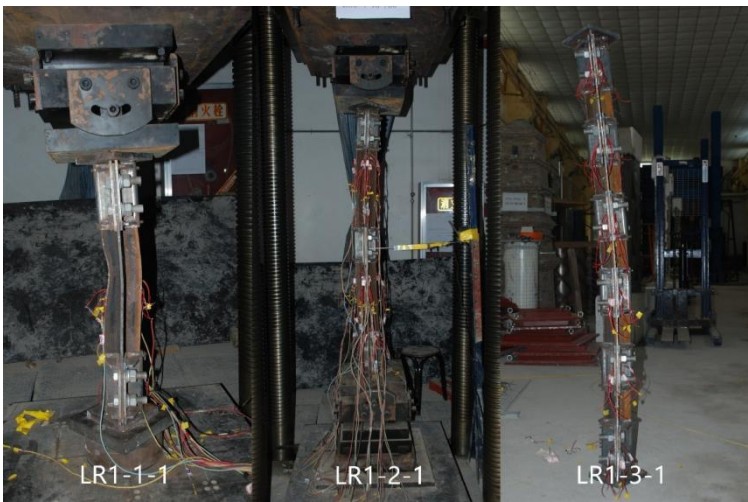

**Figure 6.** Flexural–torsional buckling failure mode.

2.4.3. Local Buckling Mode

In the experiment, it was found that when the clamps at both ends of the reinforcement steel angle are far from the end plate, the reinforcement plate and steel angle between the end plate and the clamp do not have sufficient constraints to work together, which easily forms a weak section here, and the reinforcement plate cannot play its role. As shown in Figures 3–6, this is a failed specimen. Due to the large distance between the fixture and the end plate, the steel angle here detached from the reinforcement plate. This local steel angle was not constrained by the reinforcement plate, and local buckling occurred on the weak surface. When a failure similar to Figure 7 occurs, the load-bearing capacity of the reinforced steel angle will be significantly reduced, so this failure mode should be avoided. The test piece was invalidated in the experiment, and two clamps were added at the end to conduct the test again, achieving good test results.

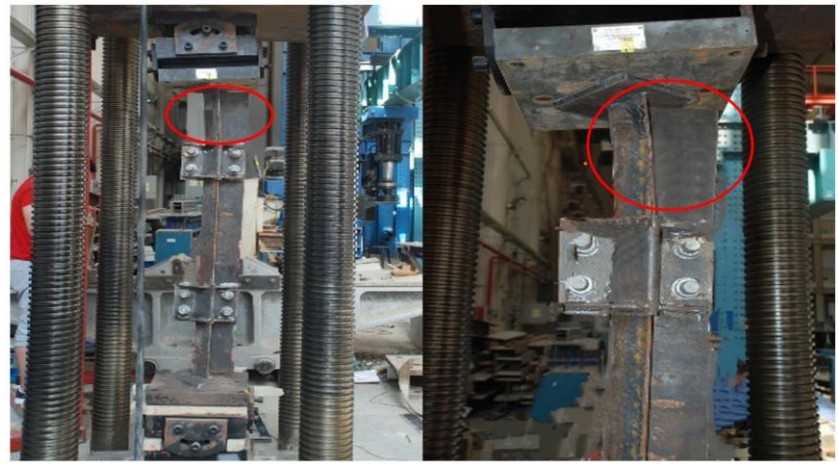

**Figure 7.** Local buckling failure mode.

## 3. Analysis of Test Results

### 3.1. Load–Displacement Curves

The load–displacement curves of the reinforced steel angle and unreinforced steel angle are shown in Figure 8. For comparison purposes, the load–displacement curves of components with the same steel angle specifications and the same slenderness ratio are plotted in the same figure. For components that experience instability and failure, the load–displacement curve can be divided into three stages: in the first stage, starting from

the origin, the curve approximately grows in a straight line until the slope of the curve changes. This stage is a linear stage, and the material plasticity has not yet developed. The specimen has no obvious lateral bending deformation, and the material is in an elastic state; in the second stage, the slope changes between the peak points of the curve, indicating a significant lateral displacement of the specimen. The second-order effect begins to become apparent, and the specimen has already buckled, but the material has not yet yielded. Therefore, the load can still rise slightly until the steel yields; in the third stage, namely the descending section after the peak, when the limit equilibrium state is exceeded, the lateral deflection of the specimen rapidly develops, and the deformation is too large to continue bearing the load, indicating the failure of the specimen.

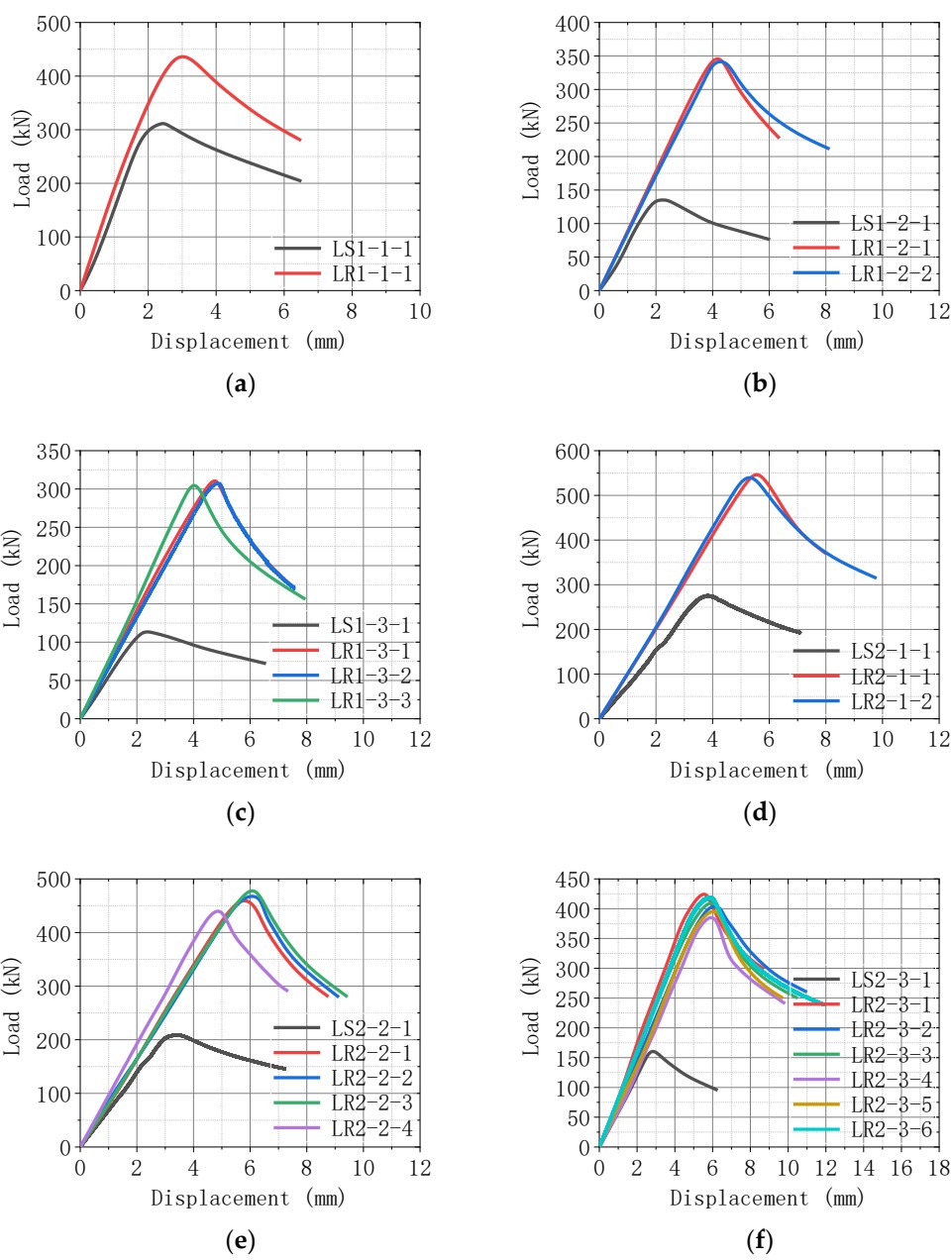

**Figure 8.** *Cont.*

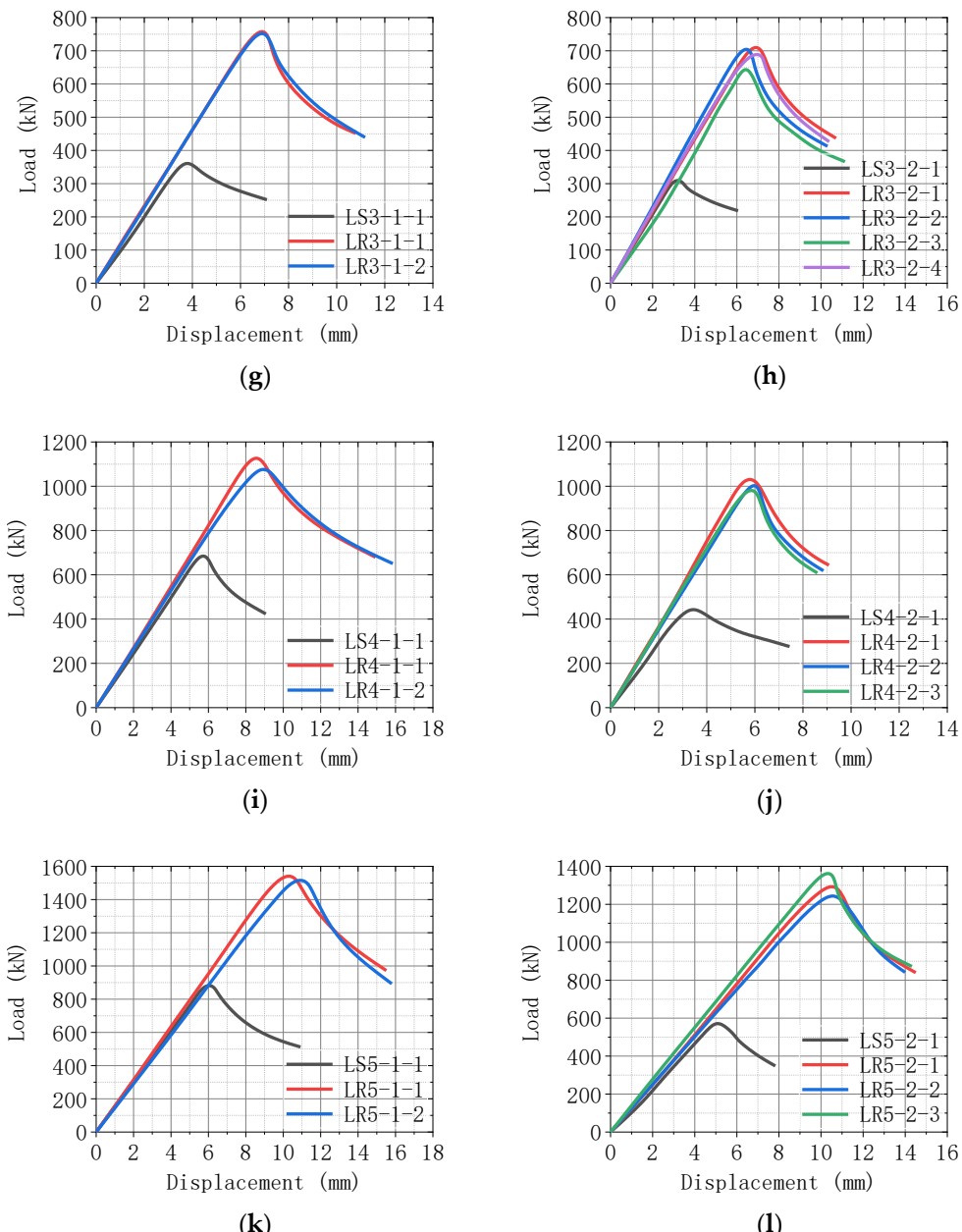

**Figure 8.** Axial load–displacement curves. (**a**) Curves of LR1-1 group; (**b**) Curves of LR1-2 group; (**c**) Curves of LR1-3 group; (**d**) Curves of LR2-1 group; (**e**) Curves of LR2-2 group; (**f**) Curves of LR2-3 group; (**g**): Curves of LR3-1 group; (**h**): Curves of LR3-2 group; (**i**) Curves of LR4-1 group; (**j**) Curves of LR4-2 group; (**k**) Curves of LR5-1 group; (**l**) Curves of LR5-2 group.

The increase in bearing capacity of the reinforced steel angle is shown in Table 3, with an increase in bearing capacity of approximately 39% to 174%, indicating a significant strengthening effect. After analysis, it is not difficult to find that the bearing capacity conforms to the following rules: Firstly, for steel angle specimens of the same specification, under the same slenderness ratio, the influence of fixture spacing and fixture form on the bearing capacity is not significant. For example, the difference in load-bearing capacity among the three groups of specimens LR1-2-1, LR1-2-2, and LR1-2-3 is within 10%, and there is no obvious distribution pattern, indicating that the fixtures used in the experiment can provide strong constraint capacity and still have optimization space. The second is that the initial aspect ratio when the steel angle is not reinforced has a significant impact on the reinforcement effect. For example, LR1-1-1, LR1-2-1, and LR1-3-1 specimens that

have slenderness ratios of 50, 90, and 110, respectively, behave with an increase in bearing capacity of 39.19% and 153.19% and 174.66%, reflecting a positive correlation between the increment in bearing capacity and the slenderness ratio. In order to better demonstrate the reinforcement effect, Table 3 calculates the critical bearing capacity of steel angle specimens under weak and strong axial instability based on AISC360-16 "American Code for Design of Steel Structures" for reference.

**Table 3.** Summary of bearing capacity.

| Retrofitted Specimens | Bearing Capacity of Retrofitted Specimens/kN | Steel Angles | Bearing Capacity of Steel Angles/kN | Capacity Enhancement | Calculation of Bearing Capacity for Weak Axis of Steel Angles/kN | Calculation of Bearing Capacity for Strong Axis of Steel Angles/kN |
|---|---|---|---|---|---|---|
| LR1-1-1 | 437.18 | LS1-1-1 | 314.09 | 39.19% | 270.87 | 378.35 |
| LR1-2-1 | 345.33 | | | 153.19% | | |
| LR1-2-2 | 339.45 | LS1-2-1 | 136.39 | 148.88% | 146.42 | 323.49 |
| LR1-3-1 | 312.26 | | | 174.66% | | |
| LR1-3-2 | 306.12 | LS1-3-1 | 113.69 | 169.26% | 97.59 | 291.74 |
| LR1-3-3 | 303.45 | | | 166.91% | | |
| LR2-1-1 | 547.03 | | | 98.65% | | |
| LR2-1-2 | 539.22 | LS2-1-1 | 275.38 | 95.81% | 228.68 | 458.37 |
| LR2-2-1 | 459.59 | | | 118.48% | | |
| LR2-2-2 | 469.27 | | | 123.08% | | |
| LR2-2-3 | 478.95 | LS2-2-1 | 210.36 | 127.68% | 157.55 | 417.03 |
| LR2-2-4 | 440.41 | | | 109.36% | | |
| LR2-3-1 | 425.01 | | | 164.67% | | |
| LR2-3-2 | 403.51 | | | 151.28% | | |
| LR2-3-3 | 410.86 | | | 155.86% | | |
| LR2-3-4 | 384.55 | LS2-3-1 | 160.58 | 139.48% | 101.96 | 373.44 |
| LR2-3-5 | 394.31 | | | 145.55% | | |
| LR2-3-6 | 418.19 | | | 160.42% | | |
| LR3-1-1 | 746.68 | | | 106.48% | | |
| LR3-1-2 | 757.13 | LS3-1-1 | 361.63 | 109.37% | 380.27 | 736.40 |
| LR3-2-1 | 709.69 | | | 130.18% | | |
| LR3-2-2 | 702.65 | | | 127.90% | | |
| LR3-2-3 | 642.44 | LS3-2-1 | 308.32 | 108.37% | 263.16 | 670.81 |
| LR3-2-4 | 687.04 | | | 122.83% | | |
| LR4-1-1 | 1126.65 | | | 63.53% | | |
| LR4-1-2 | 1071.64 | LS4-1-1 | 688.96 | 55.54% | 618.38 | 1148.19 |
| LR4-2-1 | 1032.59 | | | 132.67% | | |
| LR4-2-2 | 1002.37 | LS4-2-1 | 443.80 | 125.86% | 432.81 | 1049.17 |
| LR4-2-3 | 977.92 | | | 120.35% | | |
| LR5-1-1 | 1540.69 | | | 73.98% | | |
| LR5-1-2 | 1513.52 | LS5-1-1 | 885.56 | 70.91% | 832.78 | 1547.69 |
| LR5-2-1 | 1295.01 | | | 125.91% | | |
| LR5-2-2 | 1242.33 | LS5-2-1 | 573.23 | 116.72% | 579.64 | 1412.26 |
| LR5-2-3 | 1361.17 | | | 137.46% | | |

### 3.2. Strain Analysis

All unreinforced steel angle specimens were subjected to bending instability failure, and their load strain curve morphology is similar. LS1-1-1 and LS1-2-1 have been used as examples to illustrate, as shown in Figure 9. At the beginning of loading, the strain at all measuring points is compressive strain, and the strain values are similar, indicating that the additional bending moment of the specimen is small and the cross-sectional stress is basically uniformly distributed. Subsequently, a bifurcation point appeared in the curve, and the compressive strain at some measuring points began to decrease until they were completely in a tensile state, indicating that the specimen had already flexed and the second-order effect began to appear. Under the action of secondary bending moments,

tensile stress appeared on one side of the cross-section. It can be seen that the biggest characteristic of the load–displacement curve of the unreinforced steel angle specimen is its symmetry; that is, the strain at the two measuring points symmetrical about the $y_0$ axis is basically the same. The strain data at the measuring points SP1, SP4, SP2, and SP3 in the figure are basically consistent, indicating that the deformation of the steel angle is symmetrical about the $y_0$ axis. In addition, the slenderness ratio of the LS1-1-1 specimen is 50. The slenderness ratio of the LS1-2-1 specimen is 90, so the deflection of the LS1-1-1 specimen is smaller, and the influence of the secondary bending moment is not significant. This is manifested in the load strain curve, where SP2 and SP3 measuring points do not transition to a tensile state until the curve descends, which is in stark contrast to LS1-2-1.

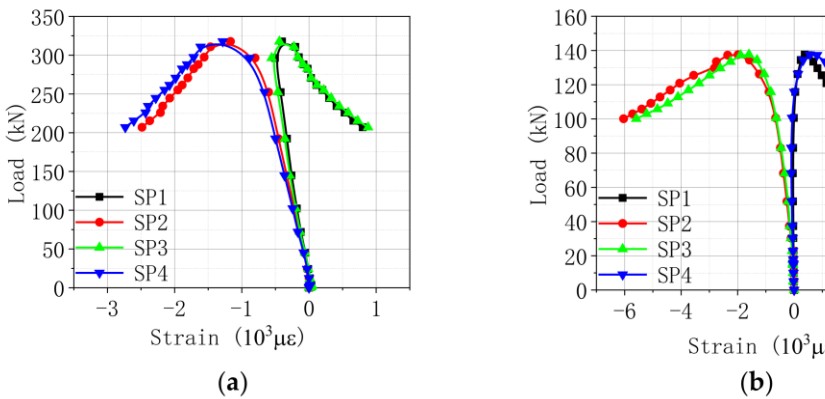

**Figure 9.** Strain analysis of steel angles. (**a**) Strain of LS1-1-1; (**b**) Strain of LS1-2-1.

As mentioned earlier, the failure modes of reinforced steel angle specimens can be divided into two types: bending torsional instability failure (such as LR1) and bending instability failure (such as LR2, LR3, LR4, LR5). There are significant differences in the load strain curves between the two failure modes, so they are explained separately. Figure 10 shows the load strain curve obtained from the measured bending and torsion specimens. It can be seen that under the bending torsional instability mode, due to the coupling of bending deformation and torsional deformation, the load strain curve no longer has symmetry. Secondly, some measuring points on the reinforced plate, such as SP6 and SP7 of LR1-2, exhibit a tensile state throughout the loading process, indicating that the reinforced plate does not directly bear vertical loads when bending and torsional instability occur.

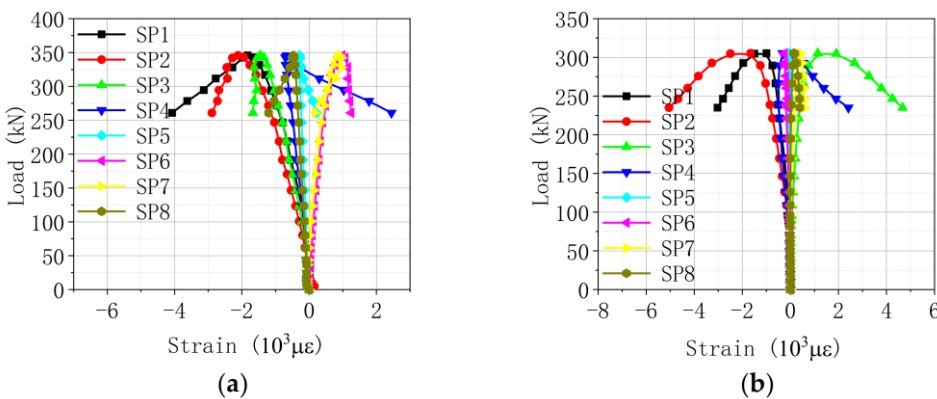

**Figure 10.** Strain analysis on retrofitted specimens under flexural–torsional buckling mode. (**a**) Strain of LR1-2-2; (**b**) Strain of LR1-3-1.

The load strain curve of the reinforced specimen under bending instability mode is illustrated using LR4-1-1 and LR4-1-2 as examples, as shown in Figure 11. The load–displacement curves of LR4-1-1 and LR4-1-2 are similar to those of the unreinforced steel

angle, and the strain values of the measuring points symmetrical about the $y_0$ axis are similar, indicating that the deformation of the steel angle is symmetrical about the $y_0$ axis. Similar to the load–displacement curve of the bending torsional instability specimen, some measuring points on the reinforced plate remained in tension during the loading process, indicating that the reinforced plate did not directly bear the vertical load when bending instability occurred.

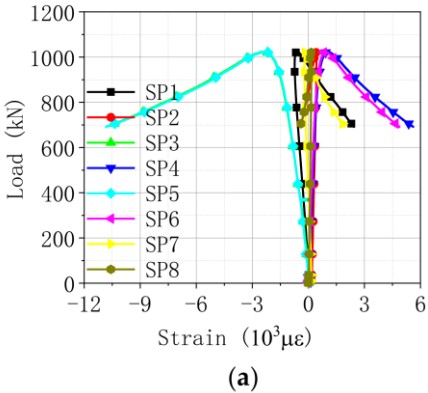 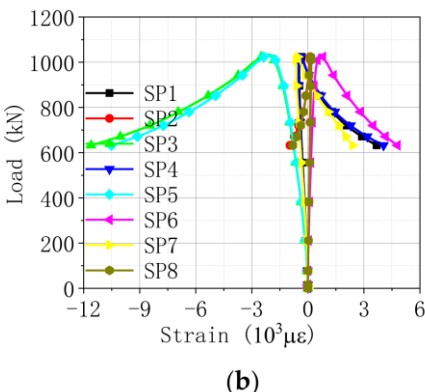

(a)    (b)

**Figure 11.** Strain analysis on retrofitted specimens under flexural-buckling mode. (**a**) Strain of LR4-1-1; (**b**) Strain of LR4-1-2.

### 3.3. Mechanism of Reinforcement Plate

Figure 12 shows the load strain curve of the measuring points on the reinforcement plate during the experiment. Figure 12a,b show the bending and torsional instability of the specimens, while Figure 12c,d show the bending instability of the specimens. From the figure, it can be seen that the SP6 and SP7 measuring points always maintain tensile strain during the loading process, even when the specimen has not buckled. It indicates that the reinforcement plate is not under pressure; that is, it does not participate in the distribution of vertical load and only balances the secondary bending moment through the squeezing force with the angle steel.

Generally speaking, for axially compressed members, the development of lateral deflection is not significant before buckling occurs, and axial displacement is mainly caused by the axial force inside the component. It is not difficult to see from Figure 13 that the slope of the load–displacement curve of the reinforced and unreinforced angle steels is very close in the ascending section, and for some specimens, it is almost identical. The slope of the rising section of the load–displacement curve represents the axial stiffness of the specimen. This indicates that the compression area of the reinforced angle steel and the unreinforced angle steel is equal and also proves that the reinforcement plate does not participate in the distribution of vertical forces.

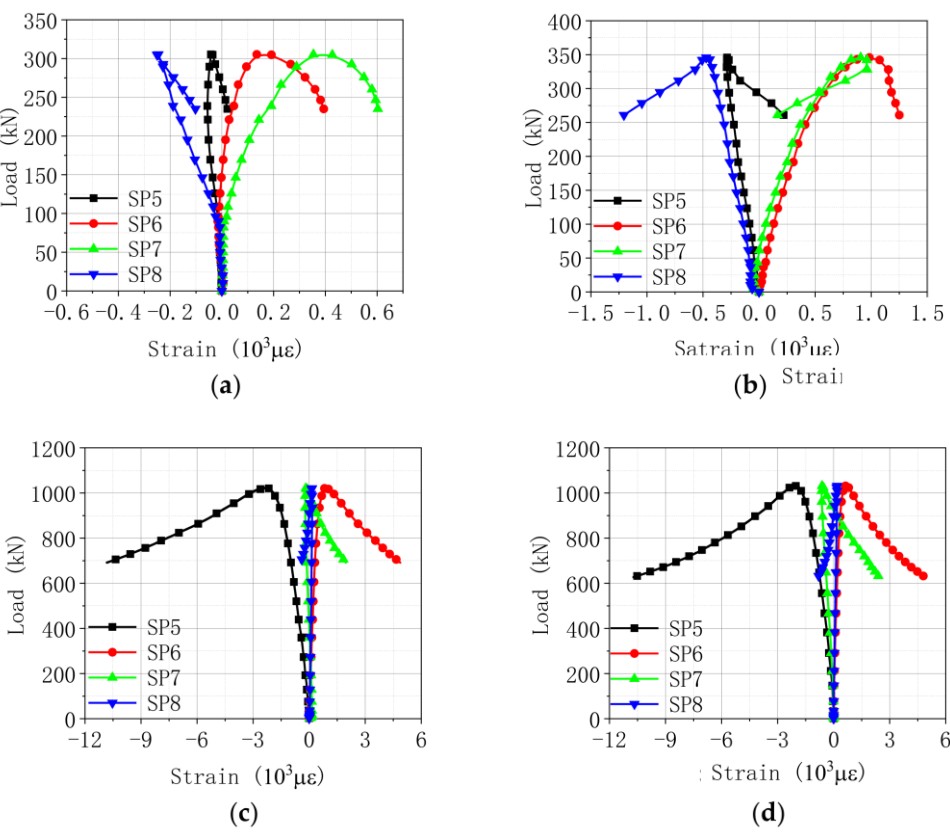

**Figure 12.** Strain analysis on reinforcement plates. (**a**) LS1-1-1; (**b**) LS1-2-1; (**c**) LR4-1-1; (**d**) LR4-1-2.

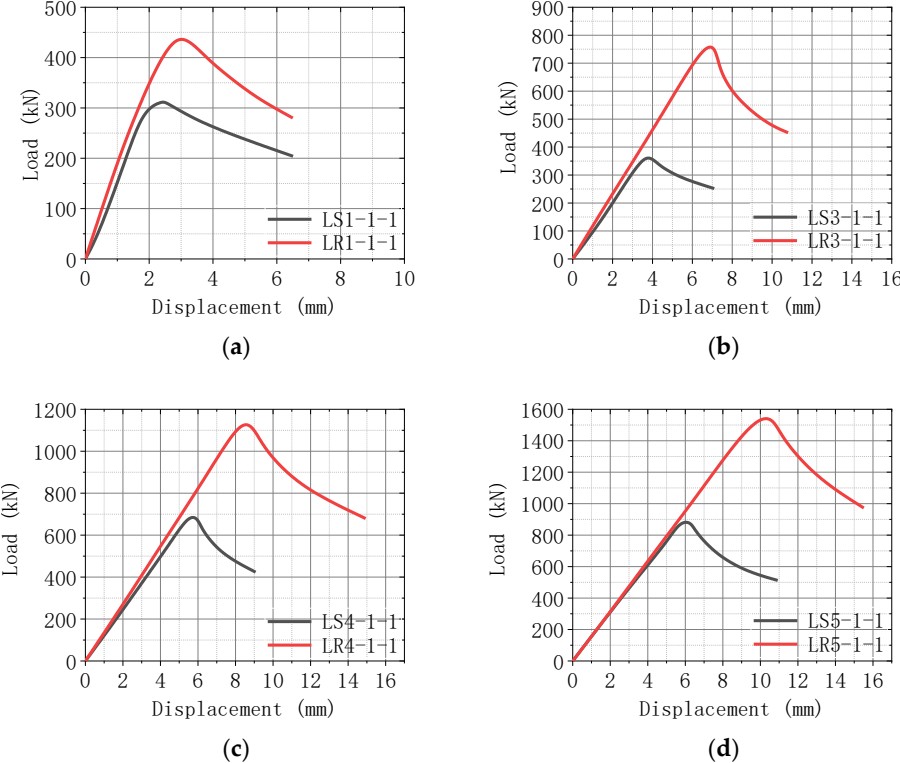

**Figure 13.** Influences of retrofitting method on steel angle. (**a**) LS1-1-1 versus LR1-1-1; (**b**) LS3-1-1 versus LR3-1-1; (**c**) LS4-1-1 versus LR4-1-1; (**d**) LS5-1-1 versus LR5-1-1.

## 4. Prediction on Ultimate Bearing Capacity

### 4.1. Bearing Capacity of Flexural–Buckling Mode

According to Section 3.3, it can be seen that it does not participate in the distribution of vertical loads and only balances the secondary bending moment through the squeezing force with the steel angle. To simplify the interaction between the reinforced plate and the steel angle, assumptions 1 and 2 are proposed, while the remaining 3, 4, and 5 are simplified methods for axially compressed members in classical stability theory.

**Assumption 1.** *There is only a normal force between the steel angle and the reinforced steel plate, ignoring the tangential friction force.*

**Assumption 2.** *The deformation of the reinforced steel plate and steel angle is completely coordinated, and the deformation of both can be expressed using the same displacement function.*

**Assumption 3.** *The pressure effectively acts along the original axis of the component.*

**Assumption 4.** *The cross-section remains plane after deformation.*

**Assumption 5.** *Small deformation assumption is adopted by assuming curvature approximately equal to the second-order differential of deflection.*

Take the isolation body of Figure 14 to establish the bending moment balance equation:

$$-EI_a w'' - P \times (w + w_0) - \int_0^z \left( q(x) + Q \right) \times (z - x) dx = 0 \tag{1}$$

where $E$ is Young's modulus; $z$ denotes the length of isolation body; $w_0$ is initial deflection at length $z$; $w$ is the corresponding deflection caused by applied load $P$; $I_a$ is the inertia moment of the angle steel, while $I_a = \iint x^2 dA$; $q(x)$ represents the normal force of the reinforcement to the steel angle; and $Q$ represents the normal force of the fixture on the angle steel.

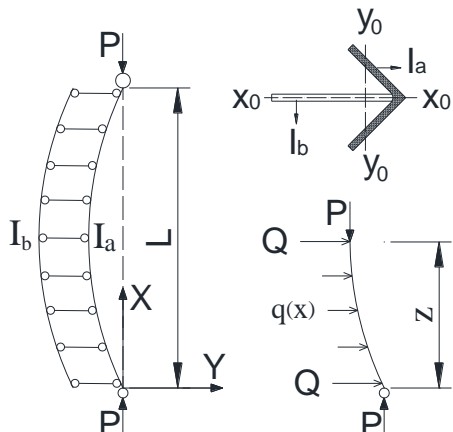

**Figure 14.** Simplified model.

Similarly, to establish a moment balance equation for the isolation body of the steel plate,

$$-EI_b w'' + \int_0^z \left( q(x) + Q \right) \times (z - x) dx = 0 \tag{2}$$

where $I_b$ is the inertia moment of the reinforcement plate, $I_b = \iint x^2 dA$.

Combine Equations (1) and (2) to obtain

$$E(I_a + I_b) w'' + P(w + w_0) = 0 \tag{3}$$

The Euler solution of Equation (3), without considering initial bending, is

$$P_{cr} = \frac{\pi^2 E(I_a + I_b)}{l^2} \tag{4}$$

where $P_{cr}$ is Euler critical load, and $l$ is the effective length of the steel angle.

According to stability theory, the deflection equation considering initial bending is

$$w + w_0 = \frac{a_0}{1 - P/P_{cr}} sin\frac{\pi z}{l} \tag{5}$$

where $a_0$ is the initial imperfection, and the initial bending approximation is represented by

$$w_0 = a_0 \sin\frac{\pi z}{l} \tag{6}$$

The deflection of the rod caused by external loads is

$$w = \frac{a_0}{P_{cr}/P - 1} sin\frac{\pi z}{l} \tag{7}$$

The bending moment shared by the steel angle ($M$) is

$$M = EI_a w'' = \frac{\pi^3 EI_a a_0}{l^2(P_{cr}/P - 1)} sin\frac{\pi z}{l} \tag{8}$$

The shear force applied to the steel angle ($Q$) is

$$Q = M' = \frac{\pi^3 EI_a a_0}{l^3(P_{cr}/P - 1)} \cos\frac{\pi z}{l} \tag{9}$$

The mid-span of the member is the most unfavorable position, and the shear force of the mid-span section is zero. The edge yield criterion is given as follows:

$$\frac{P}{A} + \frac{M}{W} < f_y \tag{10}$$

where $W$ is the cross-sectional modulus, $A$ is the sectional area of the steel angle, and $f_y$ is yield strength.

Substituting Equation (8) into Equation (10), it will obtain

$$\frac{P}{A} + \frac{\pi^2 EI_a a_0}{wl^2(P_{cr}/P - 1)} < f_y \tag{11}$$

Equation (11) is the calculation formula for the bending instability bearing capacity of reinforced steel angle.

Overall, the predicted value of the formula is close to the experimental value and numerical solution, and the main sources of error analysis are as follows: Firstly, the absolute axial pressure cannot be maintained during the test process, and the initial eccentricity of the specimen is inevitable, resulting in a relatively low bearing capacity of the measured specimen. The second is that during the installation process, there is a certain gap between the reinforcement plate and the steel angle; subsequently, in the loading process, the steel angle needs to undergo a certain degree of deflection before it contacts with the reinforcement plate, and then work together, which is equivalent to increasing the initial bending of the composite component. To consider the impact of the above two points, the prediction formula for the bending instability bearing capacity of reinforced angle steel is revised. Considering the adverse effect of installation clearance on composite components, the initial bending amplitude should be appropriately increased. After investigation, it

was found that the initial bending amplitude of L/500 is in good agreement with the experimental value. The corrected prediction results are shown in Table 4.

**Table 4.** Prediction result of simplified model.

| Specimen | Tested Result/kN | Predicted Result/kN | Tested/Predicted |
|---|---|---|---|
| LR2-1-1 | 547.03 | 503.71 | 1.086 |
| LR2-1-2 | 539.22 | | 1.070 |
| LR2-2-1 | 459.59 | | 0.973 |
| LR2-2-2 | 469.27 | 472.36 | 0.993 |
| LR2-2-3 | 478.95 | | 1.014 |
| LR2-2-4 | 440.41 | | 0.932 |
| LR2-3-1 | 425.01 | | 0.988 |
| LR2-3-2 | 403.51 | | 0.938 |
| LR2-3-3 | 410.86 | 430.30 | 0.955 |
| LR2-3-4 | 384.55 | | 0.894 |
| LR2-3-5 | 394.31 | | 0.916 |
| LR2-3-6 | 418.19 | | 0.972 |
| LR3-1-1 | 746.68 | 759.07 | 0.984 |
| LR3-1-2 | 757.13 | | 0.997 |
| LR3-2-1 | 709.69 | | 1.032 |
| LR3-2-2 | 702.65 | 687.80 | 1.022 |
| LR3-2-3 | 642.44 | | 0.934 |
| LR3-2-4 | 687.04 | | 0.999 |
| LR4-1-1 | 1126.65 | 1099.03 | 1.025 |
| LR4-1-2 | 1071.64 | | 0.975 |
| LR4-2-1 | 1032.59 | | 1.075 |
| LR4-2-2 | 1002.37 | 960.25 | 1.044 |
| LR4-2-3 | 977.92 | | 1.018 |
| LR5-1-1 | 1540.69 | 1431.94 | 1.076 |
| LR5-1-2 | 1513.52 | | 1.057 |
| LR5-2-1 | 1295.01 | | 1.056 |
| LR5-2-2 | 1242.33 | 1225.84 | 1.013 |
| LR5-2-3 | 1361.17 | | 1.110 |

### 4.2. Bearing Capacity of Flexural–Torsional Buckling Mode

For axially compressed members with a single axisymmetric cross-section, their deformation exhibits a state of both bending and torsion. For any section on the member, there is the following equilibrium equation:

$$EI_y u^{(4)} + Pu'' + Py_0\varphi'' = 0 \tag{12}$$

$$EI_\omega \varphi^{(4)} + (Pi_0{}^2 - GI_t + \overline{R})\varphi'' + Py_0 u'' = 0 \tag{13}$$

where $y_0$ is the distance from the centroid to the shear center; $u$ denotes the deflection function of the rod; $\varphi$ is the rotation angle function of the member; and $i_0$ and $\overline{R}$ can be calculated by Equations (14) and (15), respectively.

$$i_0{}^2 = (I_x + I_y)/A + y_0{}^2 \tag{14}$$

$$\overline{R} = \int_A \sigma_r(x^2 + y^2)\mathrm{d}A \tag{15}$$

where $\sigma_r$ is the residual stress distribution function on the cross-section.

The boundary conditions for both ends are hinged, and the following relationship can be obtained:

$$u(0) = u(l) = u''(0) = u''(l) \tag{16}$$

$$\varphi(0) = \varphi(l) = \varphi''(0) = \varphi''(l) \tag{17}$$

It is assumed that $u$ and $\varphi$ should satisfy the shape of a sine curve:

$$u = C_1 \sin(n\pi z/l) \tag{18}$$

$$\varphi = C_2 \sin(n\pi z/l) \tag{19}$$

where $C_1$ and $C_2$ are parameters to be determined.

The elastic solution for flexural–torsional instability can be obtained by combining Equations (12)–(19):

$$P_{y\omega} = \frac{(P_y + P_\omega) - \sqrt{(P_y + P_\omega)^2 - 4P_y P_\omega [1 - (y_0/i_0)^2]}}{2[1 - (y_0/i_0)^2]} \tag{20}$$

where $P_{y\omega}$ is bending–torsional instability critical load. $P_y$ and $P_\omega$ are bending and torsional instability critical load, which can be determined by Equations (21) and (22), respectively:

$$P_y = \pi^2 E I_y / l^2 \tag{21}$$

$$P_\omega = \frac{1}{i_0^2} \left( \frac{\pi^2 E I_\omega}{l^2} + G I_t - \overline{R} \right) \tag{22}$$

where $G$ is the shear modulus, and $I_t$ and $I_\omega$ are the torsional and warping inertia moments, respectively.

Especially for sections where the plates intersect at one point, $I_\omega \approx 0$, if the influence of residual stress can be ignored, it can be obtained by

$$P_\omega = \frac{G I_t}{i_0^2} = \frac{G(I_{ta} + I_{tb})}{i_0^2} \tag{23}$$

where $I_{ta}$ and $I_{tb}$ are the torsional inertia moments of the steel angle and reinforced plate, respectively; and $I_t$ is the sum of both and denotes the reinforced steel angle's torsional inertia moment.

It can be seen that the critical load of flexural and torsional instability can be represented by the critical load $P_y$ of flexural instability and the critical load $P_\omega$ of torsional instability. For reinforced steel angle, the influence of the reinforced plate on the moment of inertia of the steel angle section in the strong axis direction is very small, so the critical load $P_y$ for bending instability of the reinforced steel angle is approximately equal to that of the unreinforced steel angle.

Therefore, the main consideration is the contribution of the reinforced plate to the torsional instability load $P_\omega$. As shown in Equation (23), considering the effect of reinforced plates on the torsional inertia moment of the reinforced steel angle section, the cross-sectional twisting inertia moment ($I_t$) of the built-up steel angle can be determined by $I_t = I_{ta} + I_{tb}$, where $I_t$ is the inertia moment of steel angle, and $I_{tb}$ is the inertia moment of reinforcement plate. Although the above analysis is based on elastic bending and torsional instability, it reveals the mechanical mechanism of strengthening plates in improving the bending and torsional bearing capacity of reinforced steel angle; that is, by increasing the torsional moment of inertia of the reinforced section to enhance the torsional stiffness of the member, thereby improving the ultimate bending and torsional buckling bearing capacity of the member. Based on this idea, it is possible to use the torsional moment of inertia $I_t$ of the reinforced steel angle to calculate the flexural and torsional bearing capacity on the basis of the existing standard calculation methods for flexural and torsional instability. Other calculation parameters, such as the radius of gyration and cross-sectional area, are

still the same as those of the unreinforced steel angle. Using this method in conjunction with the specifications to calculate the ultimate bearing capacity of reinforced steel angle under bending and torsion buckling and comparing it with the test values measured on specimens with bending and torsion instability, the results are shown in Tables 5 and 6.

**Table 5.** Prediction result of method in GB 50017.

| Specimen | Height/mm | Effective Length/mm | Width of Reinforcement Plate/mm | Tested Result/kN | Calculated Result/kN | Tested/Calculated |
|---|---|---|---|---|---|---|
| LR1-1-1 | 790 | 1170 | 137 | 437.18 | 248.59 | 1.759 |
| LR1-2-1 | 1422 | 1802 | 137 | 345.33 | 217.87 | 1.585 |
| LR1-2-2 | 1422 | 1802 | 137 | 339.45 | 217.87 | 1.558 |
| LR1-3-1 | 1738 | 2118 | 137 | 312.26 | 198.30 | 1.575 |
| LR1-3-2 | 1738 | 2118 | 137 | 306.12 | 198.30 | 1.544 |
| LR1-3-3 | 1738 | 2118 | 137 | 303.45 | 198.30 | 1.530 |

**Table 6.** Prediction result of method in AISC 360-16.

| Specimen | Height/mm | Effective Length/mm | Width of Reinforcement Plate/mm | Tested Result/kN | Calculated Result/kN | Tested/Calculated |
|---|---|---|---|---|---|---|
| LR1-1-1 | 790 | 1170 | 137 | 437.18 | 375.71 | 1.164 |
| LR1-2-1 | 1422 | 1802 | 137 | 345.33 | 322.85 | 1.070 |
| LR1-2-2 | 1422 | 1802 | 137 | 339.45 | 322.85 | 1.051 |
| LR1-3-1 | 1738 | 2118 | 137 | 312.26 | 291.42 | 1.072 |
| LR1-3-2 | 1738 | 2118 | 137 | 306.12 | 291.42 | 1.050 |
| LR1-3-3 | 1738 | 2118 | 137 | 303.45 | 291.42 | 1.041 |

From Table 5, it can be found that the calculation results of design code GB 50017 are far lower than the ultimate bearing capacity of the reinforced angle steel measured in the experiment. During the loading process of the LR1-1-1 specimen, due to significant deformation, the end plate and reinforcement plate contacted each other, causing the reinforcement plate to directly bear vertical loads, which was not in line with the actual situation, resulting in an excessively measured ultimate bearing capacity. Except for LR1-1-1, the method in GB 50017 predicts that the bearing capacity of other specimens is about 35% lower, and the calculation formula is too conservative. The results of the American standard calculation in Table 6 are in good agreement with the experimental values, indicating the correctness of the theory of plate reinforcement under bending and torsion instability. Except for LR1-1-1, the predicted bearing capacity of all other specimens is slightly lower than the experimental value, with a maximum error of 6.67%. While leaving a certain safety margin, the bearing capacity of the reinforced angle steel is fully utilized.

## 5. Conclusions

This paper presents experimental and analytical research on retrofitted steel angle members under compression. Some conclusions can be drawn within the scope of the current study:

(1) Three failure modes of flexural-buckling mode, flexural–torsional buckling mode, and local buckling mode may occur for built-up steel angle members under axial compression. The width of the reinforcement plate has a significant impact on the failure mode. Failure of the local buckling mode can be avoided by reducing the clamp distance to the member end.

(2) The bearing capacity of the reinforced steel angles is increased by 39~174%, indicating that the reinforcement effect of the proposed non-destructive method is significant. The increment ratio of bearing capacity is positively correlated with the slenderness

ratio, reflecting that the reinforcement method is more effective for slender members. The clamp types and clamp distance behave a slight effect on bearing capacity.

(3)    By analyzing the reinforcement mechanism of reinforced angle steel, it indicates that the reinforcement plate is not subjected to axial compression; namely, it does not participate in the distribution of vertical load and only balances the secondary bending moment through the squeezing force with the steel angle.

(4)    A simplified mechanical model of reinforced steel angle members (built-up steel angles) is established under bending instability. Moreover, a design method based on existing codes is proposed to predict the flexural–torsional capacity by considering the effect of the reinforcement plate. The verification result indicates that the design method based on AISC 360-16 has better agreement with the experimental results and could be used as a basis for calculating the flexural–torsional bearing capacity of reinforced steel angles.

**Author Contributions:** Conceptualization, J.L. and Q.S.; methodology, J.L. and X.W.; validation, J.L. and B.Y.; data curation, J.L. and J.W.; writing—original draft preparation, J.L.; writing—review and editing, Q.S.; project administration, Q.S. All authors have read and agreed to the published version of the manuscript.

**Funding:** This research was funded by the National Natural Science Foundation of China, grant number 51978570.

**Institutional Review Board Statement:** Not applicable.

**Informed Consent Statement:** Not applicable.

**Data Availability Statement:** The data presented in this study are available on request from the corresponding author. The data are not publicly available due to commercial restrictions.

**Acknowledgments:** The authors greatly appreciate the help of team members of Xi'an Jiaotong University. They provided great assistance in the experimental and theoretical analysis.

**Conflicts of Interest:** The authors declare that they have no known competitive financial interest or personal relationships which may affect the work reported in this paper.

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
