# Peer review of "Experimental Study and Bearing Capacity Analysis of Retrofitted Built-Up Steel Angle Members under Axial Compression"

_applsci, doi:10.3390/app13169280_

Round 1

Reviewer 1 Report

Paper No: applsci-2464176-peer-review-v1

Title: Experimental Study and Bearing Capacity Analysis on Retrofitted Built-up Steel Angle Members under Axial Compression

In the manuscript, the authors presented an experimental study on retrofitting steel members with an angle section. The proposed retrofitting approach is constructed by using an additional reinforcement plate and auxiliary devices. Analytical investigations are also conducted to developed to evaluate the strength of retrofitted members subjected to axial compression.

In general, the manuscript is well-prepared. There are some recommendations given as follows

1.       The term “bearing capacity” could be changed to “compressive strength” of the members

2.       Is there any local failure captured in the connections between angle members and the reinforcement plate?

3.       The yield lines should be illustrated in Figs. 9-12 and yield point should be presented in load-displacement plots.

4.       Notation for axial load in Eqs. (1) – (8) should be capitalized (identical to notation if Fig. 14)

5.       How are the spacing and characteristics of connection devices considered in the proposed formulas used to predict the strength of retrofitted members? Explanations of calculations based on  GB 50017 and ACSI 360 should be provided as well.

6.       Instead of using error values, it is recommended to use the ratio of test to predicted strength in Tables 4-6.

Reviewer 2 Report

While the paper presents an economical and reasonable non-destructive reinforcement method for steel angles, the experimental results, although informative, can be considered somewhat predictable and lacking in novelty. Additionally, the paper has been assessed as being unsuitable for acceptance due to the absence of a necessary numerical description for validation and deficiencies in the theoretical model, which constitutes the innovative aspect of the paper.

1)    Table 1

Are the slenderness ratios listed in Table 1 calculated based on the total length of the specimen? Or are they calculated based on the length between knife edges? I assume that the slenderness ratio refers to the value around the minor axis, but it would be beneficial to include the value around the major axis as well.

2)    Ln 173-175

Line 175 in the paper mentions x0 and y0, but the corresponding Figure 3 does not show x0 and y0. Also, Figure 3 shows a knife edge in one direction, but the text (Ln173) describes a knife edge in two directions, making it impossible for the reviewer to determine which description is correct. Furthermore, if buckling is possible in both directions, whether the center of rotation is the same (i.e., whether the buckling length is the same in both directions) cannot be determined from this text and Figure 3.

3)    Ln262

The explanation for Figure 8 states that material plasticization has not developed. However, since Figure 8 does not include a line indicating the yield strength of the member, it is difficult to determine whether the material plasticization discussed in the paper has truly not occurred. To address this, it would be beneficial for Figure 8 to display the yield strength, calculated by multiplying the cross-sectional area by the yield stress, to provide a clearer understanding of the material behavior.

4)    Table 3

The paper suggests that the presence of the reinforcement plate and clamps enhances the flexural buckling resistance. However, it is possible that this effect is merely a transformation of the buckling mode from flexural buckling around the minor axis to flexural buckling resistance around the major axis. As previously discussed, Table 1 only presents the slenderness ratio around the minor axis, making it difficult to reach a definitive conclusion. It would be beneficial for Table 3 to include not only the experimental results but also the calculated buckling resistance values for both the minor and major axes. This addition would provide a more comprehensive analysis of the data.

5)    Section 4

The buckling resistance is theoretically derived, but the clamps that hold the reinforcement plates in place are not modeled. From the experimental results, it is concluded that the number of clamps has no effect on the flexural buckling load. But the theoretical derivation process should incorporate the clamps into the mechanical model and show that the influence of the clamps is not significant from the mathematical development.

6)    Table 4

Table 4 presents a comparison between the results of buckling load calculations based on the theoretically derived formulas and the experimental results. However, the paper does not provide the values to be assigned to the formulas, such as Ia and Ib. This omission makes it difficult for reviewers to verify the accuracy of the proposed formulas. Furthermore, it is unclear from the paper which axis of rotation, Ia or Ib, corresponds to the inertia moment. Including these crucial details in the paper would enable reviewers to better understand and evaluate the proposed formulas.

There are no major problems with the English grammar.

Reviewer 3 Report

The paper presents a novel technique to strengthen single-angle members in steel telecommunication or transmission towers. The concept is interesting and promising. However, please consider the following comments in your revised manuscript:

1- The abstract must include a brief description of the strengthening concept. It is not sufficient to say: " ... proposed a non-destructive reinforcement menthod of steel angles by avoiding ...".

2-Please revise the numbering of the specimens so that the numbering can describe the specimens better. It should show the number of clamps, free length, .... Right now, one cannot guess what is LR1-1-1 versus LR1-2-1 versus LR1-3-1 without looking at the drawings in Fig. 2. Try something along LR - angle length - slenderness rato - number of clamps - number of free spans - length of free spans - ...

3- Please revise the sentence in section 3.2, line 325-327. The two types are identified as "bending torsional instability".

4- Please rearrange the data n Tables 3 and 4, to avoid repeating information, as the second column in both tables is repeated. Consider combining the tables in one table, or presenting the data as a bar chart.

5- Please clarify why no data from specimens LR1-x-x were not presented in Table 4.

6- Please revise the "Acknowledgement" and "Conflict of Interest" declarations at the end of the manuscript. It seems you wrote the acknowledgement in the conflict of interest section, and did not provide a conflict of interest declaration.

Some sentences are very long, especially in the introduction.

Round 2

Reviewer 2 Report

Regrettably, I have observed that only a few of the appropriate corrections have been made in response to the points I raised. As it currently stands, it is challenging to assess the manuscript in its current adopted state.

Since this is a scientific paper, I strongly believe that it is essential to address peer review items 4, 5, and 6 at the very least. Properly addressing these concerns will significantly contribute to the overall quality and credibility of the manuscript. Your attention to these aspects is highly appreciated.

Round 3

Reviewer 2 Report

Thank you for your response.